# Specialist and Generalist Fungal Parasites Induce Distinct Biochemical Changes in the Mandible Muscles of Their Host

**DOI:** 10.3390/ijms20184589

**Published:** 2019-09-17

**Authors:** Shanshan Zheng, Raquel Loreto, Philip Smith, Andrew Patterson, David Hughes, Liande Wang

**Affiliations:** 1Key Laboratory of Biopesticide and Chemical Biology, Ministry of Education, Fuzhou 350002, China; zhengshanshan0428@gmail.com; 2College of Plant Protection College, Fujian Agriculture and Forestry University, Fuzhou 350002, China; 3Department of Entomology, Pennsylvania State University, University Park, PA 16802, USA; 4Center for Infectious Diseases Dynamics, Pennsylvania State University, University Park, PA 16802, USA; raquelgloreto@gmail.com; 5CAPES Foundation, Ministry of Education of Brazil, Brasilia 70040–020, DF, Brazil; 6Metabolomics Core Facility, Huck Institutes of Life Sciences, Pennsylvania State University, University Park, PA 16802, USA; pbs13@psu.edu; 7Center for Molecular Toxicology and Carcinogenesis, Department of Veterinary and Biomedical Sciences, Pennsylvania State University, University Park, PA 16802, USA; adp117@psu.edu; 8Department of Biology, Pennsylvania State University, University Park, PA 16802, USA

**Keywords:** behavioral manipulation, muscular atrophy, fungal pathogens, mandible muscle

## Abstract

Some parasites have evolved the ability to adaptively manipulate host behavior. One notable example is the fungus *Ophiocordyceps unilateralis sensu lato*, which has evolved the ability to alter the behavior of ants in ways that enable fungal transmission and lifecycle completion. Because host mandibles are affected by the fungi, we focused on understanding changes in the metabolites of muscles during behavioral modification. We used High-Performance Liquid Chromatography-Mass/Mass (HPLC-MS/MS) to detect the metabolite difference between controls and *O. unilateralis*-infected ants. There was a significant difference between the global metabolome of *O. unilateralis*-infected ants and healthy ants, while there was no significant difference between the *Beauveria bassiana* treatment ants group compared to the healthy ants. A total of 31 and 16 of metabolites were putatively identified from comparisons of healthy ants with *O. unilateralis*-infected ants and comparisons of *B. bassiana* with *O. unilateralis*-infected samples, respectively. This result indicates that the concentrations of sugars, purines, ergothioneine, and hypoxanthine were significantly increased in *O. unilateralis*-infected ants in comparison to healthy ants and *B. bassiana*-infected ants. This study provides a comprehensive metabolic approach for understanding the interactions, at the level of host muscles, between healthy ants and fungal parasites.

## 1. Introduction

Some parasites have evolved to manipulate the biting behavior of their insect’s hosts, causing infected individuals to affix themselves by their mandibles to vegetation [1]. A notable example of this is the behavior changes in ants when infected by species of fungi *Ophiocordyceps unilateralis sensu lato* (*s.l.*) [2,3]. Fungi species in this group have evolved to control the biting behavior of ants before death, an adaptive function for the fungus because it removes the infected individual from the ant colony where its development is prevented by the cleaning behavior of the resident ant [4]. Once manipulated the fungus kills the ant and uses its body to produce spores that are released to fall to the ground where they infect foraging ants. 

Previous morphological studies on this host–parasite system revealed that the muscle fibers of the head experience significant atrophy concomitant with fungal infection [5,6]. Moreover, fungal cells penetrated between the muscle fibers [6]. Because fungi are well known to secrete chemicals that affect their environment [7], a parsimonious explanation is that the observed atrophy may be due to secreted chemicals. Recent transcriptomics work supports this hypothesis, highlighting a large and dynamic secretome of the fungus during manipulation. Although transcriptomics is an important tool that provides insights into the control of insect tissues by fungal pathogens, an additional level of detail could be achieved by analyzing the changes in chemicals directly using a metabolomics approach.

Other recent studies of the metabolome of entomopathogenic fungi have been described and have also revealed aspects of host–parasite interactions [8,9,10]. In fungi, secondary metabolites are secreted through the cell wall and these play an important role in most environmental interactions [7]. A previous ex vivo metabolite study on *O. unilateralis* demonstrated that guanidinobutyric acid and sphingosine are two candidate compounds that are secreted in the presence of host tissues and may be involved in the observed morphological changes that underlie the behavioral manipulation. It was also demonstrated that *O. unilateralis* reacts differently to the brains of different ant species and when grown alongside muscles versus brains, as well as reacting differently to live versus dead tissue [11]. 

In the present study, we went beyond ex vivo studies to assay the metabolites of *O. unilateralis*-infected *Camponotus castaneus* ants at the time of manipulation. We focused on the mandibular muscles of ants since this is the most crucial tissue involved in the biting behavior that typifies this complex behavioral manipulation [6]. Since changes in the metabolome may reflect general pathology we also examined the metabolome of muscles of ants infected by the generalist fungal pathogen *Beauveria bassiana* as a positive control. The fungus *B. bassiana* can infect and kill ants (and other insects) but does not control host behavior. We used healthy ants as a negative control. We compared the metabolomes of these three groups using High-Performance Liquid Chromatography-Mass Spectrometry (HPLC-MS). We found that there were significant differences in the muscles of ants according to treatment, and the differences were most pronounced in behaviorally manipulated ants.

## 2. Results

We used High-Performance Liquid Chromatography- Mass Spectrometry (HPLC-MS) to detect the metabolite difference between controls and *O. unilateralis*-infected ants. We found a significant difference between the global metabolome of *O. unilateralis*-infected ants and healthy ants, while there was no significant difference between the *B. bassiana*-treatment ants group compared to the healthy ants. 

### 2.1. Difference between the Metabolome of Fungal-Infected Ant Muscles

We compared the global metabolome of ant mandible muscles across three treatments (Figure 1). The PCA showed that muscles of ants infected by *O. unilateralis* were markedly different from the muscles of either the positive controls (ants infected by *B. bassiana*) or healthy ants (Figure 1A,B). The clustering showed that the metabolites in the muscles from behaviorally manipulated ants (infected by *O. unilateralis*), that were biting at the time of sampling, were clearly different from metabolites in the other two groups. Moreover, the two other muscle categories (infected with the fungus *B. bassiana* and non-infected) had a less pronounced difference between each other compared to the *O. unilateralis*-infected ants’ mandible muscles. 

### 2.2. Identifying Tissues Metabolites between Behaviorally Manipulated Ants and Controls

To systematically delineate metabolite changes associated with the modulation of the biting behavior, we performed metabolic profiling of infected and healthy ants by using HPLC-QTOFMS. OPLS-DA showed a significant difference in muscles between the two treatments. These sample points, with both positive and negative ion data, were divided into two separate groups in the score plot (positive, R^2^Y = 0.997, Q^2^ = 0.919, *p* < 0.001; negative, R^2^Y = 0.990, Q^2^ = 0.964, *p* < 0.001), separating the healthy and *O. unilateralis*-infected insects (Figure 2A,B). 

To try and identify the metabolites secreted by infected and healthy ants, we used SIMCA-P + 13 to perform OPLS-DA and S-plot analysis (Figure 2B,D). We used a *p*(corr)> ±0.8 to identify 382 and 247 significantly different retention time/mass to charge ratio signals for the positive and negative metabolites, respectively.Among the metabolites that were identified by MS/MS, 31 chemical compounds were putatively identified by using Metlin Database. They included sugars, fatty acids, nucleosides, amino acids, and antioxidant chemicals. Overall, several metabolites displayed considerable differences between *O. unilateralis* and healthy ants (Table 1). The MS/MS was matched with the 23 commercial standards.

### 2.3. Different Metabolites in the Comparison between Ophiocordyceps unilateralis s.l. and Beauveria bassiana-Infected Ants

HPLC-MS/MS was performed to determine the relative levels of metabolites in mandiblemuscle infected by *O. unilateralis* ants that caused the characteristic biting behavior and muscle infected by *B. bassiana* fungi that did not. Muscle samples were clustered into two distinct groups in the OPLS-DA score plot (positive, R^2^Y = 0.980, Q^2^ = 0.905, *p* < 0.001; negative, R^2^Y = 0.985, Q^2^ = 0.935, *p* < 0.001) (Figure 3A,C). The results showed that the muscles of behaviorally manipulated ants had 455 metabolites (277 positive and 184 negative) significantly up- or down-regulated compared to the muscles of ants infected by the generalist fungus, *B. bassiana* (Figure 3B,D). The 16 putatively identified metabolites are shown in Table 2. Using commercial standards we could confirm 11 metabolites.

From S-plot analysis, we found 569 features were significantly increased or decreased in the mandible muscles of *O. unilateralis*-infected ants compared to healthy ants. We found 380 features were significantly increased or decreased in *O. unilateralis*-infected ants compared to *B. bassiana*-infected ants. Only 202 of the increased and 96 decreased ion features were shared when mandible muscles of *O. unilateralis*-infected ants were compared to mandible muscle *B. bassiana*-infected ants (Figure 4). To visualize the significantly changed mandible muscle metabolites we created a heat map using the Euclidean distance measure (Figure 5).

The identified metabolites were imported into the MetaboAnalyst online database for metabolic pathway enrichment analysis. The hypergeometric distribution method was used for analysis to calculate the *p*-value of each metabolic pathway. Enrichment analysis and topological analysis of the pathways in which the differential metabolites are located revealed a pathway with a high correlation with metabolites (*p*-value less than 0.05, influence value greater than 0.1) (Figure 6). The most relevant metabolic pathway is the purine metabolic pathway (*p* = 1.3968 × 10^−4^, impact = 0.14393).

When the metabolites from *O. unilateralis*-infected antsvs. *B. bassiana*-infected and non-infected samples were compared, we observed a notable increase of purines in terms of the levels of adenine, guanine, uracil, and hypoxanthine. Additionally, levels of adenosine, composed of adenine and ribose sugar, were significantly higher in behaviorally manipulated ants. On the other hand, adenosine monophosphate (AMP) levels were lower in *O. unilateralis*-infected ants compared with *B. bassiana*-infected samples. Sugar levels were higher in *O. unilateralis*-infected ants, with higher levels of maltohexaose, raffinose, and maltoheptaose. We identified erogothione in *O. unilateralis*-infected ants. Our results show that levels of erogothione, purines, and sugar were significantly increased after infection, indicating they could have potential role in regulating biting behavior of the ants.

## 3. Materials and Methods 

### 3.1. Camponotus Castaneus Ant-Rearing and Treatments

Insects used for the experiments were selected from three *C. castaneus* colonies (KFM1, KFM22, and KFMX), which have approximately 200 workers, obtained in South Carolina, USA in August 2014. Ants were reared in three large plastic boxes (27 cm × 19 cm × 10 cm) in the laboratory and supplied with 30% sugar-water solution and water. Colonies were maintained under a standard 12 h light, 12 h dark photoperiod cycle in 24 ± 1 °C laboratory conditions over the course of one week before the experiment. Samples were collected from there colonies. A total of 145 ants were used in the experiments described here.

### 3.2. Ophiocordyceps unilateralis s.l. Infection 

The pathogen *O. unilateralis* was obtained from infected *C. castaneus* ants in South Carolina, USA. The fungus *O. unilateralis* was grown on potato dextrose agar (PDA) medium in petri dishes. The injection of ants with the fungus was done following the protocol described by de Bekker et al. [11]. Briefly, part of the fungal colony, one-eighth of the fungal culture, which had grown to an approximately 1-cm diameter, was collected, suspended in 500 µL Grace’s medium solution, and placed in a tissue lyser for 30 s to mechanically break the colony up into small hyphal fragments. The solution of hyphal fragments was then diluted four times before use. For each colony, 30 ants were infected with *O. unilateralis*. Each ant was injected with 1 µL of fungal suspension through the intersegmental membranes underneath the forelegs. In total, 90 ants were used for the *O. unilateralis* infection. Once infected we returned infected ants to their colony and observed them for between 15 and 25 days post infection and only sampled ants that were behaviorally manipulated to bite. 

### 3.3. Beauveria Bassiana Infection

We used the generalist fungal pathogen *B. bassiana* as a positive control. It is capable of infecting ants (and many different orders of insects, besides Hymepnotera) but does not adaptively manipulate ant behavior. Conidia of *B. bassiana* were obtained from Dr. Nina Jenkins, Pennsylvania State University. The concentration of *B. bassiana* conidia solution used for the experiment was 5 × 10^9^ conidia per ml in 0.05% Tween 80. For *B. bassiana* infection, 3 µL of conidia solution were spread over the intersegmental tissue that lies under the first pair of legs, where they attach to the thorax. We used this surface infection method rather than injection because ants injected with *B. bassiana* normally die within 3–5 days because the growth of this fungus is more rapid than *O. unilateralis*. Since injection itself can occasionally lead to the death of ants, it is not always possible to determine whether injected ants died from fungal infection or from the injection procedure. For each colony, 15 ants were used for *B. bassiana* infection. Then they were moved to the large plastic boxes, fed, and placed under the same laboratory conditions as *O. unilateralis*-infected ants. Ants infected with *B. bassiana* are not behaviorally manipulated prior to death. To establish that the ants were dying we checked their behavior every 15 min from the time the first dead ants appeared. It was clear when ants were moribund and close to death because they would have pronounced difficulties in walking and their body would be curled over. We are therefore confident that we collected ants which were close to death. Collected ants were snap frozen using liquid nitrogen, before being stored at −80 °C.

### 3.4. Healthy Control Ants

For each colony, 10 ants were randomly selected and moved to a new small plastic cage and fed under the same conditions as other two treatments for 15 days before they were snap frozen with liquid nitrogen and collected.

### 3.5. Metabolite Assay

All nine cages (3 colonies × 3 treatments) were placed under a 13-h light, 11-h dark photoperiod cycle and humidity was maintained at or above 60% relative humidity with temperature at 28 °C constantly (Appendix A). We used 30 ants for the metabolite study: 10 *O. unilateralis*-infected ants, 10 *B. bassiana*-infected ants, and 10 healthy ants (Appendix A). Under a dissecting microscope, the cuticle of the head was opened, on ice, to reveal ant muscles. All the extra tissues were removed to expose the muscles. The muscles of the right side of the head were transferred into a 2-mL graduated cryogenic tube and stored at −80 °C until analysis. Only one person performed all the dissections in order to minimize variability in dissection methods. The sample extraction, metabolite separation, and detection followed the methodology described previously by de Bekker et al. [11]. Briefly, the extraction solvent system was 50/50 (*v*/*v*) methanol/water with 1 µmL of chlorpropamide as an internal standard. To collect the metabolites, we added 105 µL extraction solvent to each sample and placed this in a Precellys homogenizer (Bertin Technologies, Rockville, MD, USA) for two cycles at 6800 rpm for 30 s. After homogenization and centrifugation at 20,000× *g* (4 °C) for 15 min, 75 µL of the supernatant were transferred into a low-volume plastic auto sampler vial. Samples (5 µL) were separated by reverse phase HPLC using a Prominence 20 UFLCXR system (Shimadzu, Columbia, MD, USA) with a Waters (Milford, MA, USA) BEH C18 column (100 mm × 2.1 mm, 1.7-um particle size) and maintained at 55 °C with a 20-min aqueous acetonitrile gradient at a flow rate of 250 µL/min. Solvent A was HPLC-grade water with 0.1% formic acid and Solvent B was HPLC grade acetonitrile with 0.1% formic acid. The initial conditions were 97% A and 3% B, increasing to 45% B at 10 min, and 75% B at 12 min and holding at 75% B to 17.5 min before returning to the initial conditions. The eluate was delivered into a 5600 (QTOF) triple TOF using a Duospray^TM^ ion source (all AB Sciex, Framingham, MA, USA). The capillary voltage was set at 5.5 kV in positive ion mode and 4.5 kV in negative ion mode, with a decluttering potential of 80 V. The mass spectrometer was operated in Information-Dependent Acquisition (IDA) mode with a 100-ms survey scan from 100 to 1200 *m/z* for up to 20 MS/MS product ion scans (100 ms) per duty cycle using collision energy of 50 V with a 20-V spread. The running order of the samples was randomized to minimize the effect of any long-term changes in instrument performance.

### 3.6. Statistics Analysis

Normalization is a method which accounts for different dilutions of samples by scaling the spectra to the same virtual overall concentration [12]. Normalization using the total spectral intensity can be used for MS-based metabolomics data sets, and this reduces the effect of a single prominent component in the chromatogram such as xenobiotics or artifacts, which would not be appropriate measures of concentration [13]. Chromatograms were aligned, and features (retention time *m/z* pairs) were extracted from the LC-MS data using MarkerView and normalized by using the total area sums. The features were used for metabolomics profiling by principal component analysis (PCA) [14] and heat maps with MetaboAnalyst 3.0 online (http://www.metaboanalyst.ca/); orthogonal projection for latent structures-discriminant analysis (OPLS-DA) [15] and analysis of variance (ANOVA) were carried out with SIMCA-P+13 (Umetrics). The OPLS-DA modelled covariance (cov (t_p_,X), also known as *p*(corr)), was plotted against the chemical shift and the plot was colored accordingly.The R^2^Y and Q^2^ are the two parameters that judge the model. All data were autoscaled (mean-centered and scaled to unit variance). CV-ANOVA was used for the significance testing of the OPLS-DA model, including sum of squares (SS), degrees of freedom (DF), mean square (M), *p*-value (*p*), and standard division (SD). Scores in metabolite changes to fungi infection were compared between groups using the orthogonal partial least squares discriminant analysis (OPLS-DA) score. 

### 3.7. Metabolite Identification

Metabolites were identified by a comparison of their product ion mass spectra to those present in the online Metlin Database (https://metlin.scripps.edu/metabo_search_alt2.php). Retention times and peak areas of the metabolites of interest were extracted and automatically integrated with Peak View (AB Sciex), which was also used for quantitative review of HPLC-MS/MS data in the identification of metabolites. Each putatively identified metabolite was verified with standards if they were commercially available. Metabolite standards were prepared at a fine concentration of 10 uM in 50/50 (*v*/*v*) methanol/water and prepared fresh before use.

## 4. Discussion

In order to extend our knowledge about the biochemical changes that occur inside the muscles of insects affected by fungal pathogens, we used metabolomics to compare some fractions of the biochemical composition of muscles tissue of carpenter ants (*C. castaneus*) infected by *O. unilateralis*, which is known to induce a stereotypical biting behavior prior to host death. Using HPLC-based metabolomics analysis we discovered candidate metabolites for the *O. unilateralis*-infected ants and those infected by the generalist fungal pathogen, *B. bassiana*, which does not affect behavior, but can infect and kill ants. For *O. unilateralis*-infected ants, 31 metabolites were identified and confirmed (using their commercially available standards), including ergothioneine, hypoxanthine, and saccharopine. In addition, 14 metabolites, including niacin, glycerophosphocholine, and maltotriose (isomer), were putatively identified (without standard). 

When comparing healthy ants and *B. bassiana*-infected samples to *O. unilateralis*-infected ants, we observed a significant decrease in L-leucine of *O. unilateralis*-infected ants. Leucine is an essential amino acid and we were able to confirm its presence with the use of a commercial standard. Animals cannot synthesize leucine and bioaccumulation can occur in animal tissues, as a result of consuming organisms which produce it. Several studies have suggested that leucine is a nutrient regulator of muscle protein synthesis by activating components of the mammalian target of rapamycin signaling cascade [16,17,18,19,20]. In mammals, cancer can result in cachexia, which is a reduction in muscle mass associated with tumor growth, and leucine supplementation was shown to decrease this in rats [21]. In ants infected by the behaviorally manipulating fungus *O. unilateralis,* previous work has shown a significant decrease in mitochondria which are the power generators of the cells and the source of ATP and sarcoplasmic reticula that provide calcium for actin-myosin binding [6]. We suggest that the behaviorally manipulating fungus *O. unilateralis* may be targeting muscle cells by reducing their available energy and Ca^2+^ and depleting leucine levels to prevent muscle regeneration. 

An intriguing feature of this study was the concentration of hypoxanthine, which was significantly increased in the mandible muscles of *O. unilateralis*-infected ants compared to either healthy ants or the positive control of *B. bassiana*-infected ants. For survival and growth, certain pathogens require hypoxanthine [22,23,24] and obtain this necessary nutrition by salvaging purine from their hosts. In addition, hypoxanthine-guanine phosphoribosyl transferase, a known protease, is critical to this process [25]. Studies have reported that hypoxanthine is a catabolic product of ATP [26,27]. On the other hand, the breakdown of ATP also produces AMP and is associated with energy use in cellular processes. The identified hypoxanthine is consistent with this catabolic cycle and it was observed that AMP content decreased in metabolite abundance for *O. unilateralis*-infected ants. Based on these results, we suggest that the infection by *O. unilateralis* might therefore stimulate ATP breakdown, which is line with previous observations on a significant reduction in mitochondria in the muscles of infected ants [6]. This mechanism is likely due to an enhancement in glucose consumption by the fungus growth, which also results in an increased rate of hypoxanthine production. Additionally, Feet et al. [28] and Phillis et al. [29] reported that high concentrations of extracellular hypoxanthine induced damage to neural tissue of the cerebral cortex. We know from histological work that a distinct atrophy occurs with widely spaced myofibrils and a previous suggestion was that muscle innervation was impacted [6]. In this study the increased concentration of hypoxanthine may suggest a mechanism by which the fungus affects the motor neurons of the muscles it is targeting.

Our result also suggests that saccharopine is strongly increased in mandible muscles after *O. unilateralis* infection. Saccharopine is a precursor of lysine, which is an essential amino acid. Some studies suggest that in fungi, lysine is synthesized through the saccharopine pathway, whereas in plants, bacterium, and animals, the pathway functions in lysine degradation [30,31,32,33,34]. Lysine biosynthesis is essential for within host entomopathogenic fungal growth [35]. Specifically, saccharopine occurs as an intermediate of lysine metabolism in the alpha-aminoadipate pathway of fungi [36,37,38,39], and can be isolated from the fungal fruiting body [40]. Therefore, the observation of saccharopine accumulated in mandible muscle of the ant suggests that this metabolite is produced by fungi using the host resources. However, the host could be a source. Outside of the central nervous systems lysine is synthesized by the aminoadipate pathway. Inside the brain lysine is synthesized by the pipecolate pathway [41]. The pipecolate pathway is mediated by lysine-alpha-ketoglutarate reductase and converts L-lysine to saccharopine [42]. This suggests that the cause for increased abundance of saccharopine observed in our system may be more complex, as both the host and the parasite are putative sources. This, interestingly, also provides a window for manipulation of the host. Because neither the controls nor the *B. bassiana*-infected samples showed the presence of saccharopine it is possible that the parasite stimulates the host to produce this compound. This probable influence could be exacted via the direct manipulation of the host, the parasite-specific immunological response of the host, or otherwise. Irrespective of the source (parasite vs. host), we suggest that saccharopine may play an important role in this host–fungal pathogen interaction. More research is needed to confirm the causal mechanism of increased saccharopine abundance.

It was also notable that the relative sugar level was notably increased in the mandible muscles of *O. unilateralis*-infected ants. Sugars play important roles in non-self-recognition, and distinct sugar signatures may elicit different immune responses [43]. In the plant–fungal antagonistic interactions we know fungi can rapidly metabolize soluble sugars of the host (glucose, fructose, and sucrose) and convert them into their own metabolites [44]. On the other hand, sugar can also promote formation of fungal tissue inside the host [45,46]. We know that *O. unilateralis* uses its host cadaver as a carbon source and sequesters host metabolic stores for its own continuous spore production. This results in fungal tissue differentiation in the post-mortem ant where a carbon-nitrogen rations analysis showed by energy storage and transport [5]. In this study, the increased level of sugars also suggests that *O. unilateralis* likely produces enzymes that have sugar-related activity. 

When comparing healthy ants and *B. bassiana*-infected samples to *O. unilateralis*-infected ants, we observed a significant increase in ergothioneine of *O. unilateralis*-infected ants. Ergothioneine is a naturally occurring betaine of histidine, wshich was found at high levels in our samples. It is only synthesized by bacteria and fungi, and accumulates in human and animals tissues in cells [47,48,49,50]. A great number of studies have suggested that ergothioneine can influence a variety of animal behaviors (including the regulation of the biosynthesis precursor hercynine [51,52]), inhibit oxidative damage in cells [53,54], and scavenge free radicals [55,56]. However, the underlying scavenging reaction of ergothioneine differs from reduced glutathione [57]. Given the role of antioxidants, ergothioneine is believed to protect conidia in the life cycle between condiogensis and germination from the toxicity of peroxide. Bello et al. [58] postulated that ergothioneine has a greater impact on conidial survival and germination than on hyphal growth. We know from previous studies that at the time of manipulation the population of the fungal colony is very high, likely as much as 50% of the total tissue inside the ant [2,6]. The fungus is principally in its hyphal body stage, presenting yeast-like cells rather than hyphae. One possibility is that ergothioneine is important during what is a period of rapid cell division by the fungus. A clear understanding of the role of ergothioneine in microbes will advance our knowledge of how this thione enhances microbial virulence and resistance to the host’s defense mechanisms to avoid complete eradication [50]. 

## 5. Conclusions

In summary, this study on the metabolomics of ant muscles when infected by a specialized fungal pathogen that has evolved the ability to manipulate ant behavior highlights key biochemical pathways that are likely important for both manipulation and host exploitation. Information obtained from the metabolite study could lead to a better understanding of how *O. unilateralis* manipulates ant behavior and muscle atrophy. In this study, the relative levels of sugars, purines, and ergothioneine were significantly increased in the *O. unilateralis*-infected ants compared to *B. bassiana*-infected ants and control ants. Most of the identified metabolites may play essential roles in manipulation. However, a challenge for metabolomics is the identification and quantification of a large fraction of unknown metabolites in complex biological samples when standards are unavailable. The identified metabolites in this study cannot provide the final answer as to whether a particular metabolite causes this death grip or not.

## Figures and Tables

**Figure 1 ijms-20-04589-f001:**
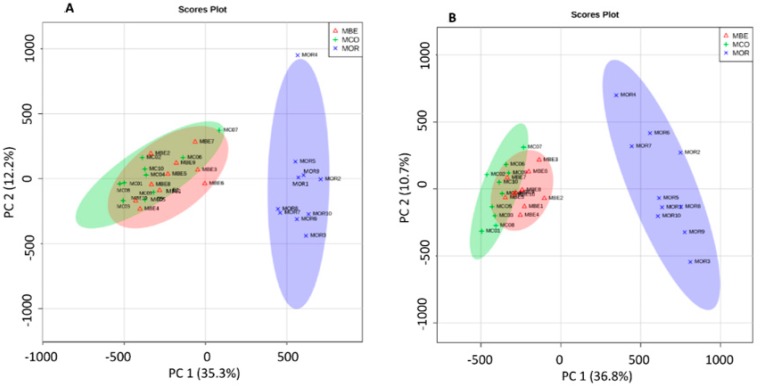
Overview of the three treatments with principal component analysis (PCA). Scores plot of a PCA model from *Ophiocordyceps unilateralis*-infected (blue, MOR), *Beauveria bassiana*-infected (red, MBE), and healthy ants’ (green, MCO) muscle samples. (**A**) PCA plot showing the clustering of three different treatment samples from positive ion MS/MS feature data. (**B**) PCA plot showing the clustering of three different treatment samples from negative ion MS/MS feature data.

**Figure 2 ijms-20-04589-f002:**
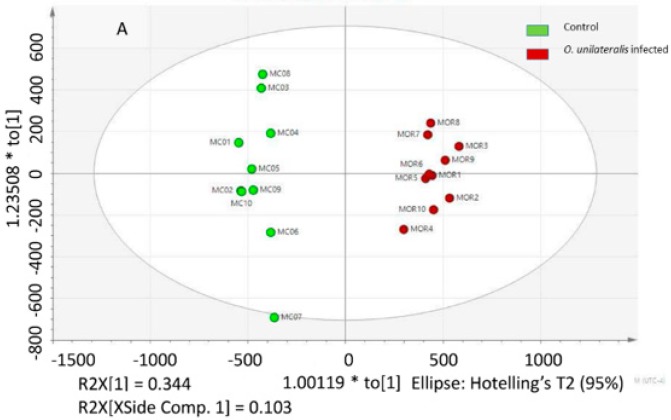
Differential metabolic profiles in muscle between healthy and *O. unilateralis*-infected ants. (**A**) Orthogonal projection for latent structures-discriminant analysis (OPLS-DA) score plot separating healthy ants (green) and *O. unilateralis*-infected (red) samples from positive ion feature data. (**B**) S-plot analysis of all *O. unilateralis*-infected samples against all healthy ants’ samples from positive ion feature data. (**C**) OPLS-DA score plot separating healthy ants (green) and *O. unilateralis*-infected (red) samples from negative ion feature data. (**D**) S-plot analysis of all *O. unilateralis*-infected samples against all healthy ants’ samples from negative ion feature data.

**Figure 3 ijms-20-04589-f003:**
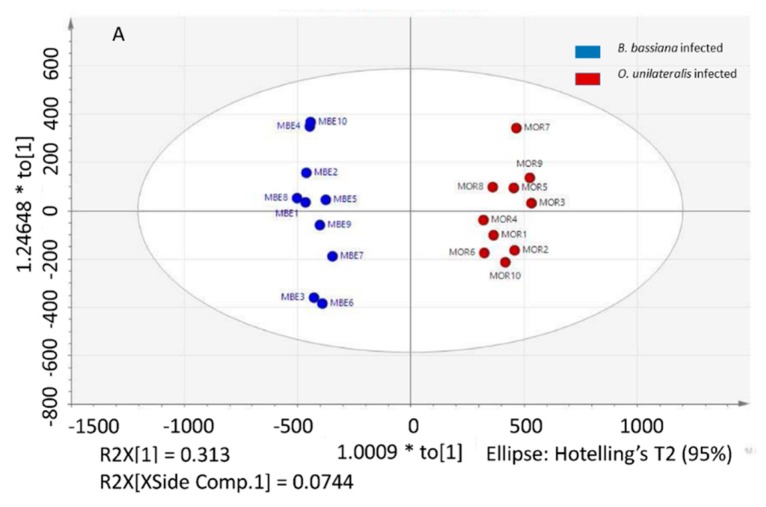
Differential metabolic profiles in muscle between *Beauveria bassiana*- and *Ophiocordyceps unilateralis*-infected ants. (**A**) OPLS-DA score plot separating *B. bassiana*-infected (blue) and *O. unilateralis*-infected (red) samples from positive ion feature data. (**B**) S-plot analysis of all *B. bassiana*-infected samples against *O. unilateralis* ants’ samples from positive ion feature data. (**C**) OPLS-DA score plot separating *B. bassiana*-infected (blue) and *O. unilateralis*-infected (red) samples from negative ion feature data. (**D**) S-plot analysis of all *B. bassiana*-infected samples against *O. unilateralis*-infected ants’ samples from negative ion feature data.

**Figure 4 ijms-20-04589-f004:**
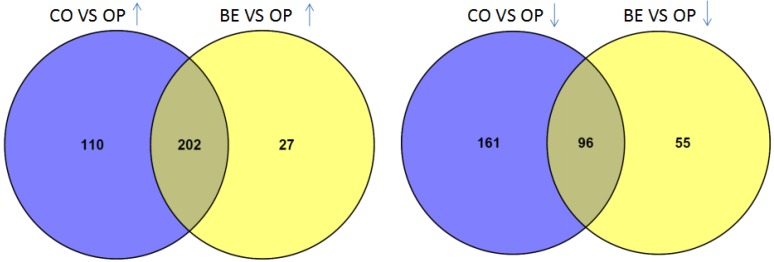
Venn diagram representing overlap between the two groups samples comparing all ion features found to be increased (↑) and decreased (↓) in *O. unilateralis*-infected mandible muscle samples based on *p*(corr) > 0.8 and *p*(corr) < −0.8, respectively (CO: control; OP: *O. unilateralis*-infected ants; BE: *B. bassiana*-infected ants).

**Figure 5 ijms-20-04589-f005:**
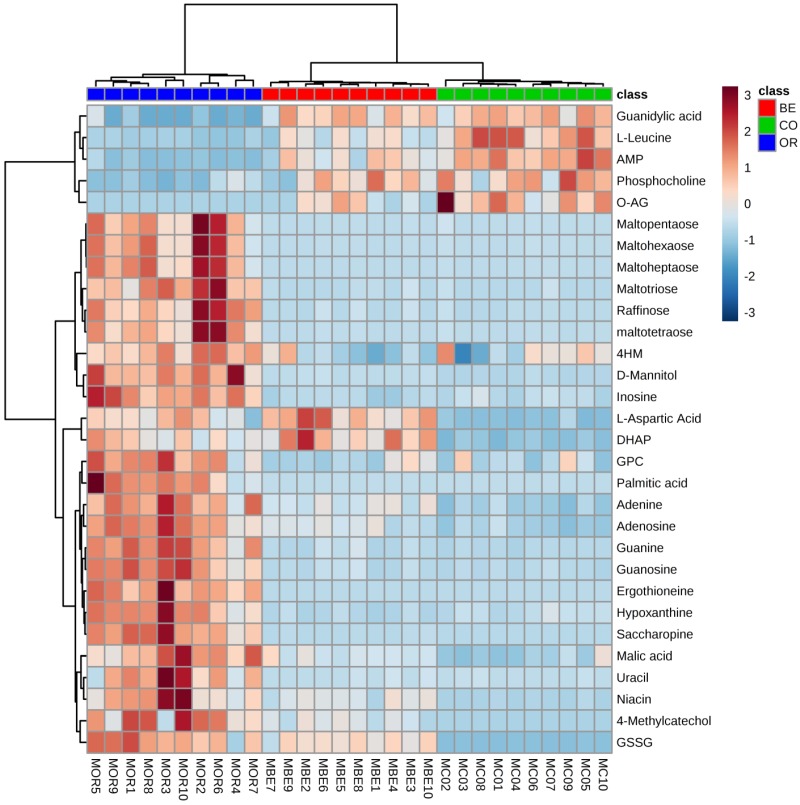
Heat maps of 30 differently identified metabolites from the mandible muscles of the three different treatment group ants: healthy ants (CO, colored green); *B. bassiana*-infected ants (BE, colored red); and *O. unilateralis*-infected ants (OR, colored blue). Each metabolite was represented by a single row of colored boxes, where columns represented different samples. Metabolite levels are scaled by colors. The red labels show high intensities levels and light blue labels mean low intensities of metabolites levels using Euclidean distance measure.

**Figure 6 ijms-20-04589-f006:**
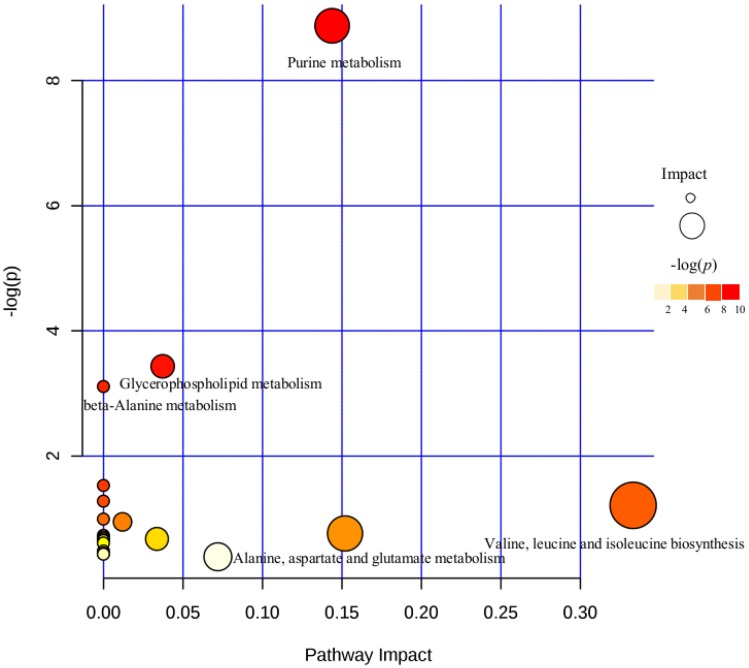
Pathway analysis for the *O. unilateralis* group vs. *B. bassiana* and the uninfected group.

**Table 1 ijms-20-04589-t001:** Compounds identified by HPLC-MS/MS analysis from healthy ants compared with *Ophiocordyceps unilateralis*-infected ants.

Compounds	*p*(corr)	*m*/*z*_Retention Time (Sample)	Sigma Cas	*m*/*z*_Retention Time (Standard)	Ionization
Guanidylic acid	−0.892268	364.0638_1.16	No	No	Positive
L-Leucine	−0.862599	132.1050_1.98	L8000	132.1019_1.84	Positive
Phosphocholine	−0.820645	184.0729_0.98	P0378	184.0717_0.95	Positive
O-Arachidonoyl Glycidol (O-AG)	−0.809626	361.2721_13.63	No	No	Positive
Phosphocholine	−0.82065	184.0733_0.98	A1752	184.0717_0.95	Positive
Niacin	0.807232	124.0395_1.45	No	No	Positive
Uracil	0.808126	113.0350_1.61	U0750	113.0357_1.45	Positive
Palmitic acid	0.814864	279.2302_13.84	No	No	Positive
Glycerophosphocholine	0.852189	258.1107_0.99	111–02−4	258.1095_0.94	Positive
Maltotriose	0.863712	505.1750_1.07	No	No	Positive
Ergothioneine	0.915323	230.0957_1.03	E7521	230.0957_1.01	Positive
Guanosine	0.928987	284.0967_1.06	G6753	284.1002_1.15	Positive
Saccharopine	0.933777	299.1281_1.00	S1634	299.1280_0.92	Positive
L-Glutathione (Oxidized) (GSSG)	0.941697	613.1566_1.08	150568	613.1733_1.03	Positive
Adenosine	0.944343	268.1037_1.06	A9251	268.1029_1.02	Positive
Adenine	0.95219	136.0630_1.08	A8626	136.0643_1.12	Positive
Guanine	0.961177	152.0574_1.08	G11950	152.0558_1.10	Positive
4-Methylcatechol	0.793135	123.0454_3.90	452–86−8	123.0457_5.83(Isomer)	Negative
L-Aspartic Acid	0.804651	132.0312_1.18	A9256	132.0417_1.06	Negative
Maltohexaose	0.804924	989.3323_1.26	M9153	989.3382_1.03(Isomer)	Negative
Maltotetraose	0.809164	665.2176_1.23	No	No	Negative
Malic acid	0.832735	133.0150_1.42	No	No	Negative
Dihydroxy acetone phosphate (DHAP)	0.845409	168.9923_1.31	D7137	168.9923_1.10	Negative
Raffinose	0.851847	503.1649_1.22	R0514	503.1649_0.97(Isomer)	Negative
GSSG	0.866878	611.1549_1.25	G4626	611.1553_1.28	Negative
D-Mannitol	0.871107	181.0722_1.20	443907(EMD)	181.0729_0.97	Negative
Guanosine	0.883834	282.0838_1.26	G6752	282.0849_1.52	Negative
Inosine	0.898411	267.0734_1.89	I4125	267.0762_1.55	Negative

**Table 2 ijms-20-04589-t002:** Compounds identified by HPLC-MS/MS analysis from *Beauveria bassiana*-infected ants compared with *Ophiocordyceps unilateralis*-infected ants.

Compounds	*p*(corr)	*m*/*z*_Retention Time (Sample)	Sigma Cas	*m*/*z*_Retention Time (Standard)	Ionization
Guanine	−0.915064	152.0570_1.09	G11950	152.0560_1.04	Positive
Ergothioneine	−0.908576	230.0964_1.01	E7521	230.0957_1.01	Positive
Guanosine	−0.88987	284.0967_1.06	G6753	284.1002_1.15	Positive
Hypoxanthine	−0.885579	137.0459_1.06	H9377	137.0461_1.03	Positive
Adenosine	−0.850788	268.1037_1.06	116833	268.1029_1.02	Positive
Adenine	−0.842341	136.0621_1.06	A8626	136.0643_1.12	Positive
4-Hydroxycinnamoylmethane(4-HM)	−0.805273	163.0735_9.68	No	No	Positive
Glycemphosphocholine	−0.803298	258.1095_0.99		258.1095_0.94	Positive
AMP	0.800485	348.0707_1.08	A1752	348.0688_1.03	Positive
Guanidylic acid	0.883806	364.0638_1.16	No	No	Positive
Maltopentaose	0.806347	827.2799_1.16	No	No	Negative
Maltoteraose	0.814304	665.2176_1.23	No	No	Negative
Maltoheptaose	0.837664	1151.3911_1.16	no	No	Negative
Raffinose	0.851338	503.1664_1.22	R0514	503.1664_0.97(Isomer)	Negative
D-Mannitol	0.855712	181.0722_1.20	443907(EMD)	181.0729_0.97	Negative
Inosine	0.904759	267.0734_1.89	I4125	267.0762_1.55	Negative

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
