# Peer review of "Specialist and Generalist Fungal Parasites Induce Distinct Biochemical Changes in the Mandible Muscles of Their Host"

_ijms, 2019, doi:10.3390/ijms20184589_

Round 1
Reviewer 1 Report
This paper explores the metabolome of the mandibles in fungal infected vs uninfected ants. The hypothesis and set up of the experiment was excellent in this study as is the use of Beauveria as a control fungi. The results look very robust even taking into account the relatively small sample numbers. Discussion is appropriate and gives good context.
I only have a few minor comments and suggestions prior to publication.
L98 infecting not infected
L120 remove "the" after All.
Please make the text in all figures bolder and larger and add global labels for ie infected, control. So that readers can see at a glance. This is especially necessary for Figures 1, 2, 3.
Figure 4 I suggest removing the muscle cartoons to make it clearer for readers. Use colours only from a colour blind friendly palette (with large contrast for visually impared or colour blind readers).
L278 Could you expand on "in our work" ? Have other studies given different outcomes ?
L374 to 379 are identical to the conclusion. Please remove.
A pleasure to review such a well planned study.
Author Response
Dear Reviewer,
Thank you for the reviewers’ comments concerning our manuscript entitled “Specialist and generalist fungal parasites induce distinct biochemical changes in the mandible muscles of their host” (ijms-578772). Those comments are all valuable and very helpful for revising and improving our paper, as well as the important guiding significance to our researches. We have studied comments carefully and have made correction which we hope met with approval. Revised portion are marked in red in the manuscript. The main corrections in the paper and the responds to the reviewer’s comments are as flowing:
Responds to the reviewer’s comments:
Point 1. L98 infecting not infected.
Response 1. We have changed infected to infecting. Please see line 98 in revised version.
Point 2. L120 Remove the after All
Response 2. We have removed “the” after All. Please see line 120 in revised version.
Point 3. Please make the text in all figures bolder and larger and add global labels ie infected, control. So that readers can see at a glance. This is especially necessary for Figure 1, 2,3.
Response3 The figure has changed.Please see revised version.
Point 4. Figure 4 I suggest removing the muscle cartoons to make it clearer for readers. Use colours only from a colour blind friendly palette (with large contrast for visually impared or colour blind readers).
Response 4 The muscle cartoons has removed and the colors has changed as suggested. Please see revised version.
Point 5 L278 Could you expand on "in our work"? Have other studies given different outcomes?
Response 5 Just change word “our results showed that” in revised version.
No, it haven't. At present, there is no relevant report on it.
Point 6 L374 to 379 are identical to the conclusion. Please remove.
Response 6 Done as suggested. Please see line 374 in revised version.
We appreciate for your warm work earnestly, and hope that the correction will meet with approval.
Special thanks to you for your good comments!

Reviewer 2 Report
Manuscript ID ijms-578772 describes a study using global metabolomics to compare the biochemical changes in jaw muscle associated with fungus-induced behavioral manipulation of ants to that of muscle tissue from healthy ants as well as ants infected with more generalist entomopathogens not shown to cause outward behavioral modification. This paper represents an original contribution to the fields of entomology and mycology and contains publishable results. It builds on the significant body of work by some of the co-authors of this study. It its current form, I recommend accept with revision. The contributions the work could potentially make would be compromised were it published without considering/addressing the following issues. I suggest you consider amending your paper to address/expand upon the issues listed below prior to resubmitting.
Main Issues:
This reviewer was somewhat concerned with the difference in treatment applications: O. unilateralis was injected into ants with 1 µl (of an unknown concentration) of inoculum in diluted Grace’s solution through the intersegmental membrane underneath the foreleg, B. bassiana in 0.05% Tween 80 was superficially applied to ants at a volume of 3 µl and concentration of 5 x 109, and negative control ants were moved into new plastic cages with no mock injections or mock superficial application.
The author’s state: “the results showed the muscles of behaviorally manipulated ants had 455 metabolites significantly up- or down-regulated compared to muscles of ants infected by the generalist fungus B. bassiana.” How do the authors know with confidence that the stress of injection in one treatment but not the other, the use of grace’s media in one but not the other, the use of tween in one treatment but not the other don’t contribute at all to these findings without controls that were mock-injected or the use of graces media in B. bassiana treated ants.
The treatments are what they are but their differences may also partially stimulate the host to produce these compounds? Perhaps I’m not understanding the treatment formulations clearly. If these differences are real then as least explore the possibility that stress from direct injections may contribute to differences.
One of the explanations for not injecting with Bb treatment was those treated with this fungus die quicker. Individual ants were treated with 1.5 x 107 conidia (3 µl of 5 x 109 /mL). That’s still an incredibly high inoculum load and the authors could have decreased this substantially as a means on slowing disease progression, perhaps even allowing for direct injections.
Which metabolites listed in Tables 1 and 2 were confirmed against an experimentally observed standard. All? If only some please note as those compounds lacking direct comparisons with commercially available standards could represent isobaric compounds and weaken the conclusions of the overall study having not experimentally validated those compounds.
Other Issues:
Line 58- other recent studies of the metabolome of entomopathogenic fungi have been described and have also revealed aspects of host-parasite interactions. Please cite:
Wrońska, A.K., Boguś, M.I., Kaczmarek, A., Kazek, M., 2018. Harman and norharman, metabolites of entomopathogenic fungus Conidiobolus coronatus (Entomopthorales), disorganize development of Galleria mellonella (Lepidoptera) and affect serotonin-regulating enzymes. PLoS One 13 (10), e0204828. Boyce, G.R., Gluck-Thaler, E., Slot, J.C., Stajich, J.E., Davis, W.J., James, T.Y., Cooley, J.R., Panaccione, D.G., Eilenberg, J., Henrik, H. and Macias, A.M., 2019. Psychoactive plant-and mushroom-associated alkaloids from two behavior modifying cicada pathogens. Fungal Ecology, 41, pp.147-164.
Line 98- infecting not infected
Line 249- misplaced period after O. in O. unilateralis
Line 289- italicize O. unilateralis
Line 306- italicize O. unilateralis
Author Response
Dear Reviewer,
Thank you for the reviewers’ comments concerning our manuscript entitled “Specialist and generalist fungal parasites induce distinct biochemical changes in the mandible muscles of their host” (ijms-578772). Those comments are all valuable and very helpful for revising and improving our paper, as well as the important guiding significance to our researches. We have studied comments carefully and have made correction which we hope met with approval. Revised portion are marked in red in the manuscript. The main corrections in the paper and the responds to the reviewer’s comments are as flowing:
Responds to the reviewer’s comments:
Main Issues: This reviewer was somewhat concerned with the difference in treatment applications: O. unilateralis was injected into ants with 1uL(of an unknown concerntration) of inoculums in diluted Grace's solution through the intersegmental membrane underneath the foreleg, B. bassiana in 0.05% Tween 80 was superficially applied to ants at a volume of 3uL and concentration of 5x109, and negative control ants were moved into new plastic cages with no mock injections of mock superficial application.
The author's state: the results showed muscles of behaviorally manipulated ants has 455 metabolites significantly up- or down-regulated compared to muscles of ants infected by the generalist fungus B. bassiana. How do the authors know with confidence that the stress of injection in one treatment but not the other, the use of grace's media in one but not the other, the use of tween80 in one treatment not the other don't contribute at all to these findings without controls that were mock-injected or the use of grace media in B.bassiana treated ants.
The treatments are what they are but differences may also partially stimulate the host to produce these compounds? Perhaps I'm not understanding the treatment formulations clearly. If these differences are real then as at least explore the possibility that stress from direct injections may contribute to differences.
Response to main issues: O. unilateralis cannot produce spores under laboratory conditions. part of the fungal colony, 1/8th of the fungal culture, which had grown to approximately 1cm diameter, was collected, suspended in 500 µl Grace’s medium solution and placed in a tissues lyser for 30sec to mechanically break the colony up into small hyphal fragments. The solution of hyphal fragments was then diluted four times before use. The effect of Grace's medium injectionon ant metabolites has disappeared within two weeks. This preliminary experimental result is also reflected in the published articles(Bekker C D , Quevillon L E , Smith P B, Fleming K R, Ghosh D, Patterson A D, Hughes DP. Species-specific ant brain manipulation by a specialized fungal parasite. BMC Evolutionary Biology, 2014, 14(1):166.).
Tween 80 is a common auxilliaries for dispersing spores. 0.05% Tween 80 is used to dispersing spores and activing B. bassiana. 0.05% Tween 80 does not affect the result of the experiment. We used this surface infection method rather than injection because ants injected with B. bassiana normally die within 3-5 days because the growth of this fungus is more rapid than O. unilateralis. Since injection itself can occasionally lead to the death of ants, it is not always possible to determine whether injected ants died from fungal infection or from the injection procedure.
Main issues: Which metabolites listed in Tables 1 and 2 confirmed against an experimentally observed standard. All? If only some please note as those compounds lacking direct comparisons with commercially available standards could represent isobaric compounds and weaken the conclusions of the overall study having not experimentally validated those compounds.
Response 2 In Tables1 and 2, m/z_ retention time(standard) column no means it does not have commercially standards, and they are putatively identified metabolites. Others with m/z_retention means they have commercially standards and identified with standards. In the conclusions and discussion we focus on the compounds that confirmed with commercial standards.
other issues:
Point 3 Line 58-other recent studies of the metabolome of entomopathogenic fungi have been described and have also revealed aspects of host-parasite interactions.
Response 2 Thank you for the reference. We cited those references in the revised manuscript as suggested. Please see revised version.
Point 4 line 98-infecting not infected
Response 4 We have changed infected to infecting. Please see line 98 in revised version.
Point 5 Line 249- misplaced period after O.unilateralis
Response 5 we have corrected it. Please see the revised version.
Point 6 Line 289 Italicize O.unilateralis
Response 6 We have changed it. Please see the revised version.
Point 7 Line 306 Italicize O.unilateralis
Response 7 We have changed it. Please see the revised version.
We appreciate for your warm work earnestly, and hope that the correction will meet with approval.
Once again, thank you very much for your comments and suggestions.
